# Sarcomas of the Larynx: One Institution’s Experience and Treatment Protocol Analyses

**DOI:** 10.3390/medicina57030192

**Published:** 2021-02-25

**Authors:** Jaromir Astl, Richard Holy, Inna Tuckova, Tomas Belsan, Miloslav Pala, Jan Rotnagl

**Affiliations:** 1Department of Otorhinolaryngology and Maxillofacial Surgery, Military University Hospital, 16902 Prague, Czech Republic; jaromir.astl@seznam.cz (J.A.); jan.rotnagl@uvn.cz (J.R.); 2Third Faculty of Medicine, Charles University, 10000 Prague, Czech Republic; 3Department of Otolaryngology, Institute of Postgradual Medical Education, 10005, Prague, Czech Republic; 4Department of Pathology, Military University Hospital, 16902 Prague, Czech Republic; inna.tuckova@uvn.cz; 5Department of Radiology, Military University Hospital, 16902 Prague, Czech Republic; tomas.belsan@uvn.cz; 6Institute of Radiation Oncology, Bulovka University Hospital, 18081 Prague, Czech Republic; miloslav.pala@bulovka.cz; 7First Faculty of Medicine, Charles University, 12108 Prague, Czech Republic

**Keywords:** larynx, laryngoscopy, sarcoma, surgery, laryngectomy, treatment, surviving

## Abstract

Soft tissue sarcomas in the head and neck are rare malignancies. They occur in this area in less than 1% of all malignant tumors. Some authors have described the development of sarcoma from the mesenchymal tissue in the larynx. The histological diagnosis of a sarcoma depends on the immunohistochemical investigation. In particularly difficult diagnoses, electron microscopy has to be used. The treatment recommendation depends on the histological type of sarcoma. We analysed and summarized data on the diagnostic criteria and therapy for sarcoma of the larynx presented in the literature. We present three new cases of laryngeal sarcoma and describe the analyses of the published diagnostic and treatment schedules of laryngeal sarcomas. We developed a treatment protocol recommendation for laryngeal sarcoma based on an analysis of literature data and case reports. This recommendation is based on histological type, staging, grading, size, and survival data.

## 1. Introduction

Laryngeal carcinoma accounts for 95–98% of laryngeal malignant tumors. Primary sarcoma of the larynx is a rare disease (occurrence below 1% of all malignant tumors). [1,2]. More than 50% sarcomas of larynx are fibrosarcomas, and chondrosarcomas are the next most frequent sarcomas in the larynx. Osteosarcomas, liposarcomas, undifferentiated pleomorphic sarcomas, synovial sarcomas, and rhabdomyosarcomas are very rare [2]. Laryngeal leiomyosarcoma is extremely rare [3]. Laryngeal osteosarcoma is certainly rarer than leiomyosarcoma in the same localization [4]. A rare case of moderately differentiated squamous cell carcinoma (SCC) larynx associated with osteocartilaginous metaplasia of the adjacent stroma similar to osteosarcoma is described in the literature [5].

The first two documented examples of laryngeal myofibroblastic sarcoma (low-grade form) in the world literature were described by MacGregor et al. in 2002 as a rare new subtype in the laryngeal area [6].

Low-grade myofibroblastic sarcoma (LGMS) is an atypical myofibroblastic tumor with fibromatosis-like features with preference predominantly in the head and neck area. LGMS generally was described mostly as a malignant sarcoma (low-grade type) with a considerable tendency to recur with low metastatic potential [7,8]. Only isolated cases of laryngeal low-grade myofibroblastic sarcoma have been described in the world literature [9,10,11]. Another very rare subtype of laryngeal sarcoma is undifferentiated epithelial sarcoma. This rare type of sarcoma was described in the sphenoid sinus, maxillary sinuses, and once on the vocal fold. We have only seen some case reports about this laryngeal sarcoma.

Myofibroblastic sarcomas (MFS) or myofibrosarcomas are very rare. They are identified by the structure of the cells and are characterized by special immunohistochemical markers: desmin, vimentin, alpha-smooth muscle actin. The challenging histological diagnosis of sarcoma depends on immunohistochemical investigation. In controversial cases, laryngeal sarcomas diagnosis has to be substantiated by electron microscopy [4]. 

There is a minimum of information in the literature regarding recommended therapeutic protocols for laryngeal sarcoma. The new paper by de Juan Ferré et al. (2021) on clinical guideline of management of soft-tissue sarcoma [12] is certainly worth mentioning.

In the present work, the diagnostic-therapeutic procedures were analyzed in a review of published cases of malignant sarcoma of the larynx.

## 2. Case Reports 

### 2.1. Case 1

A 63-year-old man, smoker, presented to our clinic with complaints of shortness of breath and dysphonia that had been ongoing for 3 months. This patient was referred to the Otorhinolaryngology (ENT) outpatient department with edema of the right vocal cord with a history of hoarseness. He denied any alcohol consumption.

A laryngeal examination revealed an exophytic mass in the right vocal cord. There was fixation of the right vocal cord and a tumor (size 12 mm × 15 mm × 12 mm) arising from the anterior commissure and ventral part of vocal cord.

Laryngoscopy showed the tumor of the vocal cord, the mobility of the right vocal fold had reduced with excursion of the vocal cord by the tumor mass, but the mobility was 2 mm more in the lateromedial direction. The vocal cord on the left side was completely functional. See Figure 1. 

Direct microlaryngoscopy was performed under general anesthesia with a representative biopsy taken from the lesion. The result of the histopathological examination was myofibroblastic sarcoma of the larynx (MFS).

We performed a partial laryngectomy—extended chordectomy with partial resection of the laryngeal ventricle (Morgagni’s sinus), partial resection of the perichondral layer of the thyroid cartilage, and a resection of the anterior commissure 3 mm on the left vocal cord. The resection was done with a CO_2_ Laser (8 W), and the section of the margins of the resection was histologically tumor free (frozen section and full investigation).

Radical surgery was sufficient for the treatment. We followed-up the patient, who has been without signs of recurrence or persistence of the disease for 36 months.

### 2.2. Case 2

A 67-year-old man presented to our ENT department with complaints of shortness of breath and dysphonia that had been ongoing for 2 months. This patient was referred to the ENT outpatient department with a growing tumor on the right side of the larynx. Histology of a biopsy sample verified a benign tumor. Because the biological behavior of the tumor was described as aggressive growth, a new biopsy was performed. The sample was analyzed, and histology showed a suspicious fibrosarcoma. He had not been smoking and denied alcohol consumption. Respiration was difficult.

A laryngeal examination revealed an exophytic mass in the right vocal cord and anterior commissure. The mobility of the right vocal cord was maintained, and there was a tumor of approximately 12 mm × 15 mm × 17 mm arising from the commissure and ventral part of the vocal cord. 

Direct microlaryngoscopy was performed under general anesthesia with a representative biopsy taken from the lesion. The result of the histopathological examination was myofibroblastic sarcoma of the larynx. The laryngoscopy showed the tumor of the vocal cord, the mobility of the right vocal fold was with reduced excursion of the vocal cord by the tumor mass, but the mobility was more than 2 mm in the lateromedial direction. The vocal cord on the left had complete function.

Given the respiratory distress, an urgent tracheostomy with general anesthesia was performed.

CT scans showed the transglottic location of the tumor mass being 40 mm × 22 mm × 16 mm into the base of the epiglottic cartilage. No infiltration to the cartilage was evident. No lymph nodes were described, as the largest lymph node was not over 5 mm. See Figure 2.

We performed a total laryngectomy. The tumor in the larynx was completely obstructed by the laryngeal lumen. Revision of the lymph nodes on both sides was performed, and no evidence of enlargement was detected. A histopathological diagnosis of myofibroblastic sarcoma of the larynx (MFS / LGMS) was performed.

The next step involved beam therapy because the tumor margin was near the surgical board of the sample. The dose was 66 Grays. The patient is still alive, and no evidence of recurrence of the tumor or persistence of the disease had been detected to date.

### 2.3. Case 3

A 80-year-old man presented to our ENT department with the following health problems: shortness of breath and growing of neck tumor mass ongoing for 2 months. This patient was referred to the ENT outpatient department with a large tumor on the jugular region and thyroid gland. He had stopped smoking 40 years ago. He denied alcohol consumption. He had urethral cancer 10 years ago.

A laryngeal examination revealed an exophytic mass with spread into the larynx, thyroid, and soft tissue. The infiltration of the chest aperture was evident, and multifocal metastases in the lung were detected. Multifocal skin metastases were also detected. There was fixation of the right vocal cord and a tumor of approximately 12 mm × 15 mm × 12 mm arising from the commissure and ventral part of the vocal cord. See Figure 3. Direct microlaryngoscopy was performed under general anesthesia, with a representative biopsy taken from the lesion.

An urgent complicated tracheostomy for respiration distress was done. Death occurred 20 days after tracheostomy. We waited for the results of rare histology for 3 weeks. The definitive result of histology, a rare finding, was laryngeal undifferentiated sarcoma, which not announced until after the patient’s death. The immediate cause of death according to the autopsy protocol was thrombotic embolism of the pulmonary artery branch. 

The definitive result of the histopathological examination was undifferentiated epithelial sarcoma of the larynx (UNEP/UDES). 

Table 1 shows a summary of the immunohistology of the published cases and tree new cases [3,13,14].

Table 2 presents the clinical report of staging, grading, and invasion criteria of laryngeal sarcomas.

Table 3 presents the clinical symptoms and treatment of laryngeal sarcomas MFS (LGMS) and UNEP (UDES).

## 3. Discussion 

The sarcomas are uncommon malignant tumors of the larynx. Some reports describe an incidence of soft tissue sarcomas in adults in the head and neck area below 5%. Laryngeal sarcomas are less than 1% of all malignant laryngeal tumors generally [13,15].

Sarcoma of the larynx commonly presents as hoarseness. Laryngoscopy shows a painless submucosal mass. The tumor may grow fast, with dyspnea and/or stridor following in the short period of time.

Sarcomas are described with a wide range of histological results and clinical activity from relatively slow-growing tumors (chondrosarcoma) to hostile tumors (undifferentiated epithelial sarcoma). This undifferentiated epithelial sarcoma (our Case 3) was loco-regionally destructive and had great metastatic ability.

The early symptoms of the presented cases of sarcoma were hoarseness, dyspnea, or suffocation. These have been described in many publications as well. The growth potential of sarcomas is high. Sarcomas may grow in any part of the larynx but are most often located in the vocal cords and/or cricoid cartilage [16].

We support these data with three new published cases. Our group of three patients indicates that the etiology of laryngeal sarcomas is still unclear. Their incidence is more common in men and in older age.

The treatment of LGMS is still not precisely determined. Most papers supported surgical treatment—excision of the localized tumor, radiation therapy of the primary area, or a use of both treatment methods.

Most patients were treated surgically. Either by local excision or some broad excision (laryngectomy). Only a few cases have been reported where patients were treated with additional radioactive treatment or chemotherapy (yet their role is controversial). Many authors recommend a broad wide local excision of LGMS and subsequent careful dispensarization [7,8].

If the sarcoma grows slowly, the tumor is often small in size and has a late recurrence. In tumor recurrence, the tumor does not report increased cellularity, proliferation, or increased number of atypia compared to the primary tumor. In contrast, however, one case of metastatic sarcoma exhibiting these features has been described in the literature.

In our two cases we used the wide (radical) surgical therapy despite the size of the tumor mass, age, and histology type of sarcoma. In the case with near-tumor margin we added radiotherapy.

Covello et al. published case of laryngeal LGMS (69-year-old female) surgically treated by supracricoid partial laryngectomy with cricohyoidoepiglottopexy (without block neck dissection). No further additional therapy was given. One year after surgical treatment, the patient is free of tumor recurrence/persistence [10]. Our two cases were in line with this case report.

One case of total laryngectomy for LGMS has been reported in the literature. For our one case, it was also necessary to perform a total laryngectomy [6,15]. The results and 3-year disease-free interval are satisfactory for this approach.

The analysis of published data and our experience supported the following treatment protocol recommendation.

### 3.1. Diagnosis

Indirect laryngoscopy to provide evidence of laryngeal status and decision of type of larynx obstruction.CT, MRI scan show size, spread, and metastases of tumor mass.Exact diagnosis of sarcoma by biopsy under a laryngoscope needed.Histopathology.

Complete investigation by an experienced pathologist and extensive immunohistochemistry tests are necessary. The poor cases for any histological subtypes of laryngeal sarcoma were associated with poor data of immunochemistry markers for laryngeal sarcoma diagnostic set.

### 3.2. Treatment

Surgery. Surgical therapy must be radical total laryngectomy or less radical laryngeal resection. The surgical margins are very important for grading of laryngeal sarcomas in general.Radiotherapy. Beam therapy depends on the oncological protocol including staging, grading, histological type of sarcoma, and radicality of surgery if it has been done.Chemotherapy. For aggressive histological subtypes, chemotherapy (combination of vincristin, doxorubicin, cyclophosphamide/iphosphamide, mesna, and etoposide) was used. The effect of chemotherapy did not influence the survival data [12,17].

### 3.3. Follow-up for Laryngeal Sarcoma

Indirect laryngoscopy every 4 months.MRI once yearly.Direct laryngoscopy if necessary.

In our group of three cases, two patients were treated surgically (1× total laryngectomy, 1× partial laryngectomy), and one patient died of tumor generalization. One of our patients received a full dose of radiotherapy.

We did not perform block neck dissection because no pathologically altered cervical lymph nodes were seen on CT and ultrasound. According to the literature, 88–90% of sarcomas (soft tissue, head and neck area) do not have metastases in the neck area [1]. The accurate diagnosis of laryngeal sarcoma depends on the biopsy and histological examination (immunohostochemistry at the same time). There is still no standardized staging system for laryngeal sarcomas and their histological subtypes [10,13].

The staging system of soft tissue sarcoma described Ryan [18,19] involves a classification based on histological type including the immunochemistry of the tumor, grading of tumor cells, status of surgical margins, perineural and lymphovascular and cartilage invasion. Ryan described the staging system for soft tissue sarcoma as a prognostic indicator [17]. This system utilizes tumor size, invasion of tumor into cartilage (bone), vessels, nerves, neck metastases and presence of distant metastases, and histology grading of malignancy. The tumors in this paper were classified retrospectively. The American Joint Committee on Cancer (AJCC) 2010 classification was used [20]. Cates (2019) reported on a proposal for a new classification in their paper “Staging soft tissue sarcoma of the head and neck: Evaluation of the AJCC 8th edition revised T classifications” [21].

Baker published a staging system (based on grading) according to the recommendation of AJCC/UICC classification [18]. The UICC classification [22] described the classification for soft tissue sarcomas (different to rhadbomyosarcoma) in two levels:

Level 1
Metastatic type—evidence of far metastasesLocalized type—tumor is evident in the germinal area only

Level 2
Kaposi sarcomaDermatofibrosarcomaFibromatosis (desmoid)Sarcomas growing from meninges, brain, parenchyma of organs (excluding mammal sarcomas, and angiosarcoma is excluded for aggressive growing)

The soft tissue classification is used for head and neck sarcomas. [19,20,22].

For the sarcoma (head and neck), a division into stages of disease has not yet been established.

The laryngeal sarcomas were classified using this recommendation in this paper, which supported the aim of the authors of this paper. We need more information and cooperation on an international basis for a better classification of sarcomas in the head and neck region. Improving the classification can be lead to more exact and stronger evidence for modification of the treatment protocol and improving patient outcomes in the future. The treatment and prognosis can be divided according to the biological behavior of sarcomas in three subtypes. Our data for laryngeal sarcomas and the literature overview data can be used for dividing laryngeal sarcomas in three groups depending of the malignant potential of each histological subtype of laryngeal sarcoma:

Group 1. Subtype 1 Laryngeal Sarcomas with Good Prognosis
Chondrosarcoma [23]Myofibroblastic sarcoma [9,14]Leiomyosarcoma [3,15]Hemangiosarcoma (angiosarcoma) [15]

Group 2. Subtype 2 Laryngeal Sarcoma with Moderate Survival Data
Kaposi sarcoma [24]Fibrosarcoma [15]Liposarcoma [24]

Group 3. Subtype 3 Laryngeal Sarcoma with Poor Prognosis
Rhadbomyosarcoma [24]Pleomorphic non-differentiated sarcomaSynovial sarcoma [25]Alveolar soft-tissue sarcomaUnclassified/non-differentiated sarcoma [15]

The general recommendation for treatment and planning protocol for laryngeal sarcoma unfolds according to the size and biologic behavior.

### 3.4. Surgery

In the literature, the type of surgery recommended is (total or partial) laryngectomy depending on the size of tumors. We can use the endoscopy approach, vertical partial laryngectomy (laryngofissure), or total laryngectomy for large tumors with infiltration of a large area of laryngeal structures. On the other hand, for any other histological types (chondrosarcoma) we can use reduced surgery (de-escalated surgery). The literature described that because many of these tumors are pedunculated, show less tendency to infiltrate the surrounding structures, and metastasize later than laryngeal carcinomas, they may remain operable for a considerably longer period after diagnosis compared to squamous cell carcinoma [15]. Italian authors treated 29 patients with laryngeal nonsquamous malignancies surgically: 10× total laryngectomy and 19× partial laryngectomy; 16 patients were alive and free from disease after treatment [26].

From our group of three patients, two patients underwent surgical treatment (1× total laryngectomy, 1× partial laryngectomy), with one receiving a full dose of radiotherapy as adjuvant therapy because of the proximity of the surgical margins at the tumor border.

Surgical treatment is the main treatment method in the planning of therapy for laryngeal sarcoma. Preservation of the larynx is feasible, as most laryngeal sarcomas are detected early. According to the literature, 10–12% of sarcomas (soft tissue, head and neck area) develop early metastases in the neck area [10,16,24]. In our group, the surgically treated patients did not have nodal or far metastases generally. We used the radical surgery treatment because the sarcomas were diagnosed early. Some authors have described fibrosarcoma, which occurs more often. This is also a poorly differentiated type. This tumor metastasizes in less than 25% of cases [25]. Elective block neck dissection is in general not required.

In our study, one surgically treated patient with a neck mass (enlargement lymph nodes, evident on US, CT, NMR) had a block neck dissection (10%), which proved to be a metastasis in the neck area.

### 3.5. Radiotherapy (RT)

Radiotherapy in the treatment protocol for head and neck sarcomas has shifted significantly over the last 30 years. The literature indicated that sarcomas demonstrate considerable radio-resistance. By contrast, some authors cited radiotherapy as a substantial supplement in the treatment protocol of soft-tissue sarcoma to reduce the incidence of local recurrence [23,24,25].

The main indications for postoperative radiotherapy (recommended in the literature) are high-grade sarcoma, positive surgical edges, large sarcoma (over 5 cm), and tumor recurrence. In our study, adjunctive postoperative radiotherapy was given in one patient with high-grade sarcoma, expecting to improve local control.

### 3.6. Chemotherapy

The outcome of chemotherapy on different sarcomas of the larynx has not been verified statistically. None of the published cases were given chemotherapy. On the other hand, we have introduced non-surgical therapeutic oncological protocols including chemotherapy and radiotherapy leiomyosarcoma [3].

We accept that neoadjuvant chemotherapy and radiotherapy have been reported for rhabdomyosarcoma only [16]. Under these therapeutic conditions, therapy failure was documented in one case, and the patient died. This patient underwent tracheostomy because of suffocation only. He died 21 days after the biopsy because of generalization of the sarcoma into the skin, lymph nodes, and lung (Level 2, T4b, Stage IV). In most cases of soft tissue sarcomas (head and neck region), the grade of tumors and margin status is important for local sarcoma control. Most papers described the formation of distant metastases as the most important survival indicator in this heterogeneous sarcoma group. Dissimilar histology types and Tumour, Node, Metastasis (TNM) classifications of Malignant Tumors may also affect the prediction of soft tissue sarcomas.

On the other hand, a chondrosarcoma is often well differentiated (histopathological classification) and has better prognosis [16]. Usually, the survival rate for head and neck sarcomas is worse than for limb sarcomas. According to the literature data, 5-year survival of patients with soft-tissue sarcoma of the head and neck ranged between 32%–75%. Table 4 summarized the review of literature on head and neck sarcoma.

Due to the rarity of laryngeal sarcomas, no survival data have yet been available. The absence of a large group of laryngeal sarcomas is due to the fact that laryngeal sarcoma is an especial tumor.

A surgical solution is the therapeutic method of choice. Early diagnosis of laryngeal sarcoma makes it possible to choose surgical treatment with the protection of the larynx in the treatment protocol. Adjuvant radiotherapy (postoperative) is applied to high-grade sarcoma and/or positive surgical edges. The prognosis is comparatively favorable when compared with sarcoma arising from other anatomical areas. A report from the year 2020 by de Juan Ferré et al. provides a current view of the diagnostic-therapeutic model for soft-tissue sarcoma. The diagnostic-therapeutic management should be executed by multi-disciplinary team. MRI/CT of the tumor-area and biopsy is compulsory before the initiation of treatment. Radical surgery with tumor-free tissue edge is the treatment pillar for the localized lesion. RT is recommended in big-size, deep, high-grade sarcomas, or after marginal resection not suitable for re-excision. Perioperative chemotherapy should be used for high-risk sarcomas of the extremities and trunk-wall. In the case of oligometastatic disease, patients should be recommended local therapies [12].

## 4. Conclusions

Based on the staging of sarcoma [19] and our previous experience, we would like to present our modified treatment protocol for laryngeal sarcoma for expert discussion. It is designed based on the experience of one institution and a literature data analysis:

Level 1 Localized type—surgery

Level 2 Metastatic type—no surgery

### 4.1. Surgery

T1 tumor less than 2 cm in main diameter—larynx preservation procedures limited and/or partial laryngectomy if necessary.

T2 tumor over 2 cm but less than 4 cm in main diameter—partial or total laryngectomy.

T3 tumor over 4 cm localized type into larynx—total laryngectomy.

N0—no evidence of nodal metastases, no block lymph nodes dissection.

N1—evidence of lymph node metastases, block (radical) neck dissection.

M0—no evidence of far and organ metastases.

### 4.2. No Surgery

T4a tumor of the brain wraps, brain, orbit, masticatory muscles, and face skeleton.

T4b tumor growing into brain, surrounding the carotid, infiltrating prevertebral space or central nervous system through perineurial growing.

N—any/arbitrary

M—any/arbitrary

Adjuvant radiotherapy (postoperative) is only applied to high-grade sarcoma and positive surgical edges.

Neoadjuvant chemotherapy and/or radiotherapy is recommend for rhadbomyosarcoma only.

## Figures and Tables

**Figure 1 medicina-57-00192-f001:**
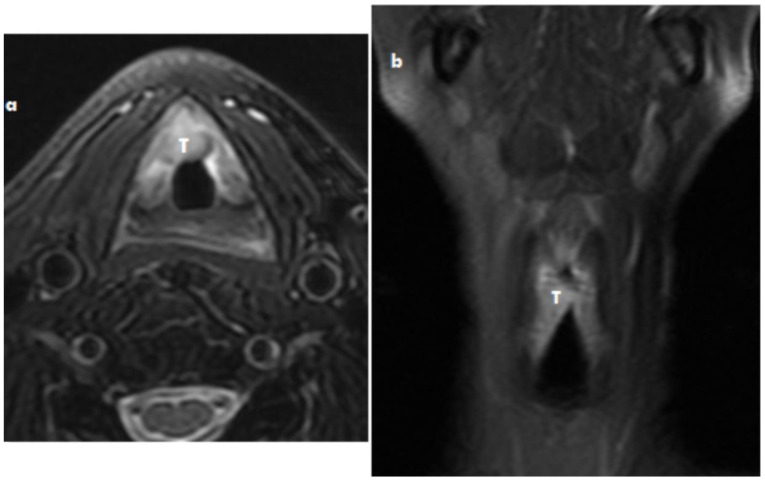
Myofibroblastic sarcoma MRI axial scan (**a**) and coronar scan of the larynx (**b**) T2 imaging with depression of fat signal: The tumor mass (T) on the right part of the larynx and anterior commissure without spread into thyroid cartilidge.

**Figure 2 medicina-57-00192-f002:**
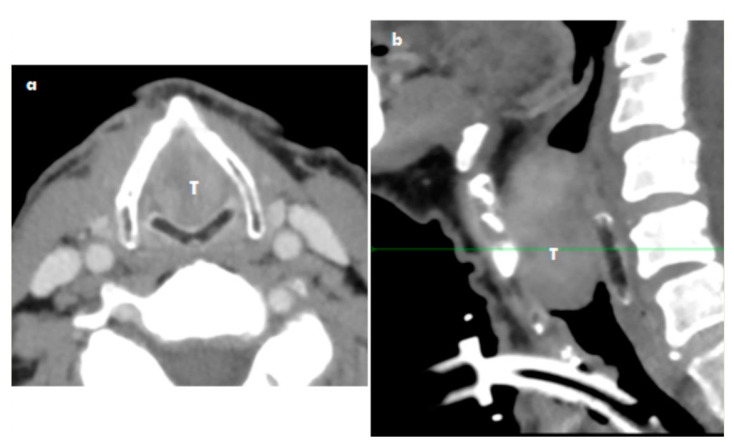
Myofibroblastic sarcoma CT axial scan (**a**) and coronar scan of the larynx (**b**) use iodine contrast: The tumor mass (T) with comletly obstruction of larynx without spread of tumor into thyroid and cricoid cartilidge. The trachestomy (trachestomy tube) was done before the CT (reason emergency suffocation).

**Figure 3 medicina-57-00192-f003:**
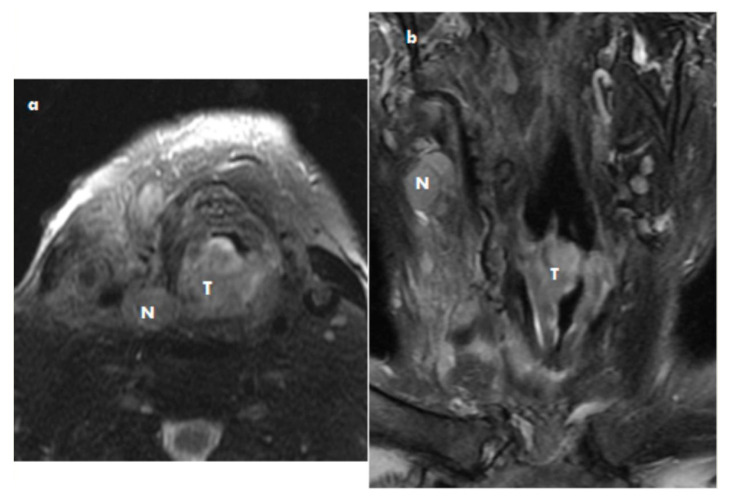
Epithelial sarcoma MRI axial scan (**a**) and coronar scan of the larynx (**b**) T2 imaging with depression of fat signal: The tumor mass (T) on the right part of the larynx and nodal metastases (N).

**Table 1 medicina-57-00192-t001:** Summary of immunohistology of published cases and tree new cases.

Marker	Case 1 63-Year-Old Man	Case 2 67-Year-Old Man	Case 3 80-Year-Old Man	Astl et al. 2019 [3]	Kordač et al. 2014 [13]	Lobo et al. 2016 [14]
Histology type	MFS (LGMS)	MFS (LGMS)	UNEP (UDES)	LMS	MFS (LGMS)	MFS (LGMS)
Mitotic activity	20 mitoses/ 10 HPF	5–6 mitoses/10 HPF	4–5 mitoses/10 HPF	High		More 10 mitoses/10 HPF Proliferation index 10%
Ki67	25–30%	12%		5–10%		
Vimentin	+++	+++	+++	None		+++
Aktin	+++			Sarcomeric actin ++		
Myogenin	None	None	None	+++		
Beta-katein	Cytoplasma membrane					
Desmin	-	-	-	+++	+	Fordesmin -
myoD1	-		None	-		
S100	-	-	-		-	-
ALK	-				-	-
CK AE1/3	-	-	-	-		AE5/6,18 -
P 63	-			-		
EMA	-	-				-
Calponin	Partial +				+	
SMA	Partial +	Partial +	-	Focal +		
EGFR epithelial	-					
Synaptofyzin			-			
H caldesmon				Negative in tumor Positive in vessels	Negative in tumor	
Chromogranin A			-			
CD31	-					
CD34	-	-	-		+	-
CD99			-		+	-
HMB45			-			
CD68						-
GFAP						-

Explanations of abbreviations: MFS: Myofibroblastic sarcoma of the larynx, LGMS: Low-grade myofibroblastic sarcoma of the larynx, UNEP/UDES: Undifferentiated epithelial sarcoma of the larynx, LMS: Leiomyosarcoma of the larynx, HPF: High Power Field, Ki67: Marker Of Proliferation (the name is derived from the city of origin Kiel, Germany), myo D1: also known as Myoblast Determination Protein 1, S-100 protein: marker of neuroektodermal differentiation, ALK: Anaplastic lymphoma kinase, CK AE1/3: an antibody cocktail being generally positive in cancers of epithelial origin, CK 5/6, 18: antibody positive in cancers of epithelial origin, P63: TP63 transcription protein, EMA: Epithelial Membrane Antigen, SMA: Smooth Muscle Actin Monoclonal Antibody, EGFR: Epidermal Growth Factor Receptor, CD31: endothelial cell adhesion molecule, CD34: hematopoietic progenitor cell antigen, CD99: cell surface glycoprotein, HMB45: Human Melanoma Black, CD68: Cluster of Differentiation 68, GFAP: Glial Fibrillary Acidic Protein. The crosses (plus) + to +++ correspond to the expression of the marker expression intensity. The minus—means the absence of marker expression.

**Table 2 medicina-57-00192-t002:** Clinical report of staging, grading, and invasion criteria of laryngeal sarcomas.

Marker	Case 1 63-Year-Old Man	Case 2 67-Year-Old Man	Case 3 80-Year-Old Man	Astl et al. 2019 [3]	Kordač et al. 2014 [13]	Lobo et al. 2016 [14]
Histology	MFS	MFS	UNEP	LMS	MFB	MFB
Grading FNCLCC	Grade 2	Grade 2	Death	Grade 2(+3)	Not published	Not published
pTNM	pT1pN0pM0	pT3pN0pM0	pT4pN3pM1 (lung, skin)	pT3pN0pM0	pT1N0M0	pT1N0M0
Perineural invasion	Negative	Negative	Present	Negative	Not published	Not published
Lymphangio invasion	Negative	Negative	Massive	Negative	Not published	Not published
Cartilage spread	Negative	Moderate, tumor cells invasion in perichondal layer	Massive Tumor going through	Negative	No	No
Distant metastases	No	No	Multinodal	No	No	No
Surgical margins	No, tumor close to margin	Negative	Positive in hole biopsy sample	Negative	Negative	Negative

Explanations of abbreviations: FNCLCC—Federation Nationale des Centres de Lutte Contre le Cancer.

**Table 3 medicina-57-00192-t003:** Clinical symptoms and treatment of laryngeal sarcomas MFS (LGMS) and UNEP (UDES).

Criteria	Case 1 63-Year-Old Man	Case 2 67-Year-Old Man	Case 3 80-Year-Old Man
Hoarseness	Yes	Yes	Yes
Dyspnea Tracheostomy	Yes	Yes	Yes
Lymph nodes enlargement	None	None	Yes
Computer Tomography (CT) imagine of the tumor	Yes	Yes	Yes
Biopsy preop histology	Yes	Yes	Yes
Radical surgery	Partial Laryngectomy	Total laryngectomy	None
Histology	Myofibroblastic sarcoma	Myofibroblastic sarcoma	Undifferentiated epithelial sarcoma
Beam therapy	None	Yes 66 Gy	None
Disease free interval (month)	36	39	0
Survival (month)	36	39	1

**Table 4 medicina-57-00192-t004:** Review of literature on sarcoma of the head and neck.

Author	Pathologic Type	No. Patients	5-Year OS
Farr, 1981 [27]	Head and neck region	285	32%
Littman et al., 1983 [28]	Head and neck region	32	75%
Farhood et al., 1990 [1]	Head and neck region	176	55%
Le et al., 1997 [29]	Head and neck region	65	56%
Dudhat et al., 2000 [30]	Head and neck region	72	60%
Our group	Larynx sarcoma	3	66%

OS = overall survival rate.

## Data Availability

Not applicable.

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
