# Peer review of "Sarcomas of the Larynx: One Institution’s Experience and Treatment Protocol Analyses"

_medicina, 2021, doi:10.3390/medicina57030192_

Round 1

Reviewer 1 Report

 Sarcomas of the larynx: one institution experience, treatment 2 protocol analyses.

The authors aimed to present three new cases of laryngeal sarcoma. The design of the study project is extremely simple but the description of rare malignant laryngeal histotypes is always interesting.

Abstract: The authors should report that in particularly difficult diagnoses electron microscopy has to be used.

Introduction: Second para: The reported data on the relative frequency of these rare laryngeal neoplasms need to be reconsidered. It will be useful for the authors to consider the Discussion section of DOI:10.1080/00016480410017288. Laryngeal osteosarcoma is certainly rarer than leiomyosarcoma of the same site (DOI:10.1016/j.amjoto.2004.11.007).  Last para:  I agree that immunohistochemical investigations are necessary to distinguishing laryngeal sarcomas from other spindle-cell tumors. On the other hand, the authors should consider that IHC can produce ambiguous or inconclusive results when the tumor cells lack specific reactivity. In controversial cases, laryngeal sarcomas diagnosis has to be substantiated by electron microscopy. In the last para, the investigation end-points should be defined and described in detail.

Discussion: A part of the first paragraph is redundant (see Introduction). Suggested diagnostic approach should include, in selected cases, electron microscopy. In treatment sub-section a clarifying note must be included regarding the management of the cN0 neck. Regarding surgical treatment, the results of DOI: 10.1016/j.amjoto.2007.02.007 should be commented. The paragraph regarding laryngeal sarcoma staging according to AJCC 2010 classification is highly debatable… A paragraph should be devoted to critically analyze more in depth strengths and weaknesses of the present investigation. A paragraph should also be dedicated to commenting in more detail the potential role in diagnostic, therapeutic and prognostic terms of the preliminary evidence obtained.

The Conclusions are definitely too drastic coming from a very exploratory study. I suggest to the authors a more moderate approach pending confirmation from multi-institutional prospective studies.

Author Response

ANSWER FOR REVIEWER ATTACHED IN THE WORD DOCUMENT

Reviewer 2 Report

Congratulations to the authors for reporting case reports of such rare malignancies. Since the disease is extremely rare, is understandable that large and homogenous cohort are arduous to be collected. Most of the current literature is based on case reports, small series, and reviews.
I understand the difficulties in collecting data, reporting the survival outcomes from a single Institution experience.
However, I find that the presented paper does not provide any additional evidence to the current literature. Thus several published articles reported extensive review of the literature, from the diagnosis to the treatment strategies.
As a suggestion to the authors, I would recommend performing an English proof-review by a native speaker.
Moreover, since the staging system for head and neck sarcoma is a matter of International debate, I would add a subchapter on sarcoma staging limits/concerns. This article could be interesting as a reference 10.1002/hed.25701.

Author Response

(The authors gave the same response as above.)

Reviewer 3 Report

Sarcomas of the head and neck are very challenging topic and is very useful to share the decision making. Please revise your english (line 150 distar)

Author Response

(The authors gave the same response as above.)

Reviewer 4 Report

The authors describe a new protocol to assess and treat laryngeal sarcoma. The manuscript is of some interesting. I agree with the authors' opinion (lines 196—197) that the appropriate protocol for patients with laryngeal sarcoma is needed. However, please note that plagiarism must be avoided.

Major revisions:

#1:

My concern is that about 25 % of the sentences (except the title, authors’ affiliation and References section) seem to have been copied from published text. Apparent “copy and paste” sentences are found in the manuscript. Even with a citation, copying published text is sometimes thought as plagiarism.

For example,

lines 255—258 “Because many of....cell carcinoma.” ;

lines 260—264: “Surgery is the...metastasis early” ;

lines 266—268: “, which is .... not required.” ;

lines 276—281: “an important adjunct....high-grade tumors.” ;

lines 297—300: “In general, ... 32% to 75%.” ;

lines 305—310: “that sarcoma of....anatomical sites.” :

The above sentences are very similar to the published article (Liu CY, et al. J Chin Med Assoc. 2006 Mar;69(3):120-4. doi: 10.1016/S1726-4901(09)70189-3 ).

Further, the table 4 of the manuscript is similar to the table 2 of  the article.

Also,

lines 146—155: “Although the.....showing these features.”

The above sentences are very similar to the published article (Petr Kordac, et al. ACTA MEDICA 2014; 57(4):162–164. http://dx.doi.org/10.14712/18059694.2015.82 ).

#2: The literature review of the authors is not enough. Before establishing new protocol of laryngeal sarcoma,  the existed guidelines for laryngeal cancer and/or head and neck sarcoma should be reviewed. For example, the protocol for laryngeal cancer might be found in NCCN guidelines (https://www.nccn.org/professionals/physician_gls/pdf/head-and-neck.pdf ).

Also, the management protocol for sarcomas (including head and neck sarcoma) might be found in SEOM clinical guideline (de Juan Ferré A, et al. Clin Transl Oncol. 2021 Jan 6. doi: 10.1007/s12094-020-02534-0. ).

By doing the literature review, the author might clarify “what is known” and “what is unknown”.

Further, there are many reports of laryngeal sarcoma exist. Please clarify the novelty of the manuscript with appropriate literature review.

Author Response

(The authors gave the same response as above.)

Round 2

Reviewer 1 Report

The paper has been sufficiently improved by suggested changes. The article should be carefully re-read in search of typographical errors. 

Author Response

Thank you. We performed a second careful reading of the manuscript. 

Reviewer 2 Report

I thank the authors for making the required changes. The article can bepublished now

Author Response

(The authors gave the same response as above.)

Reviewer 4 Report

Thanks for considering all my suggestions. And, additional revisions are needed.

#1:

Apparent “copy and paste” sentences are found in the manuscript, again.

In lines 361—367: the authors described

“MRI/CT of the ... for local therapies.”

The above sentences are very similar to the abstract of the reference #12. Even with a citation, copying published text is likely to be thought as plagiarism.

#2:

lines 137—138:

What is “Embolia thrombotica ramorum segmentalium aliquot a. pulmonalis utrque”? Please explain here.

#3:

Please review the whole manuscript again to see if there are misspellings. For example,

line 280: “Pleomorfic” might be changed to “Pleomorphic”.

line 360: “sarcom” might be changed to “sarcoma”.

Author Response

Thank You very much. 

#1 In line 361-367 edited the text ( green color)

#2 in line 137-138 - specified in English language

#3  corrected the typos in the text